# Complexation of Gold(III) with Pyridoxal 5′-Phosphate-Derived Hydrazones in Aqueous Solution

**DOI:** 10.3390/molecules27217346

**Published:** 2022-10-28

**Authors:** Natalia N. Kuranova, Daniil N. Yarullin, Maksim N. Zavalishin, George A. Gamov

**Affiliations:** Department of General Chemical Technology, Ivanovo State University of Chemistry and Technology, 153000 Ivanovo, Russia

**Keywords:** gold(III), hydrazone, pyridoxal 5′-phosphate, stability constant, speciation

## Abstract

Today, complexes of gold(I) and gold(III) are recognized as promising drugs for the treatment of bacterial infectious diseases and oncological diseases, respectively. It is of interest to broaden the area of potential use of gold(III) compounds to the pathogenic microorganism as well. The first step towards the development of new antibacterial drugs based on Au^3+^ complexes is the study of their stability in an aqueous solution. The present contribution reports on the investigation of gold(III) complexation with five hydrazones derived from a well-known biologically active compound, pyridoxal 5′-phosphate (one of the aldehyde forms of the B_6_ vitamin). The complex formation in aqueous solutions was confirmed by mass spectrometry and fluorescent spectroscopy. The stoichiometric composition of the complexes formed and their stability constants were determined using a UV–Vis titration method. The complexes are quite stable at physiological values of pH, as the speciation diagrams show. The results of the paper are helpful for further studies of gold(III) complexes interaction with biomacromolecules.

## 1. Introduction

The discovery of penicillin and other antibiotics ushered in a new era that is probably now coming to an end. Antibiotic resistance among pathogenic microflora is increasing to alarmingly high levels worldwide [1]. New mechanisms of resistance are emerging and spreading all over the world, threatening the ability to treat common infectious diseases. More and more infections, such as pneumonia, tuberculosis, blood poisoning, gonorrhea, and foodborne illnesses, are becoming more difficult and sometimes impossible to treat because antibiotics are less effective. In this regard, alternative means of combating pathogens are actively sought [1]. The use of compounds of various metals, particularly gold, is among these alternatives. Gold has two main oxidation states, +1 and +3. While the Au^+^ complexes have been synthesized in abundance, and their antimicrobial and antifungal properties have been thoroughly studied (including well-known compounds used for other medical purposes, e.g., auranofin, an antiarthritic agent [2]), the gold(III) complexes, on the contrary, are mainly investigated as antitumor agents based on their similarity to platinum(II) preparations [3,4,5,6]. A small number of Au^3+^ compounds have been tested for efficacy against pathogens (see, e.g., reviews [7,8] and papers [9,10,11,12,13,14]).

In our opinion, the potential of gold(III) compounds has not been fully disclosed. From the analysis of the literature data, it can be concluded that there are two, at least, main mechanisms of the biological action of gold. First, metal has a genotoxic effect associated with the binding of coordination compounds to DNA (both intercalation [3,4,6,15,16,17,18] and covalent binding in grooves [5,19,20]) or with the cleavage of DNA caused by the formation of free cationic radicals from N-donor ligands in the presence of Au^3+^ ions [21] or the unwinding of the DNA double helix of DNA [15]. Second, it may cause the inhibition of thioredoxin reductase, which leads to oxidative stress and apoptosis. Although thioredoxin reductase is considered a target mainly in the treatment of oncological diseases, it can be crucial for the survival of pathogenic microorganisms [22,23].

The ligand helps the gold cation to permeate the cell membrane. Several ligands can be used in gold(III) complexes, particularly hydrazones, a versatile class of compounds that are known for their strong metal-chelating properties [24,25] and are widely applied in catalysis, the dye and pigment industry, as chemosensors and intermediates in organic synthesis [26], and in medicine [27]. Of particular note among hydrazones are the derivatives of pyridoxal or pyridoxal-5-phosphate (B6 vitamers)—biologically active compounds—in the synthesis and study of which we have accumulated some experience (see, e.g., [28,29,30,31,32]). These hydrazones are low-toxicity and membranotropic agents [33]. Having carried out a preliminary evaluation of the potential antibacterial activity using the PASS Online software [34] (the results are given in Appendix A), we found that vitamin B6 hydrazones, which also contain residues of furan, thiophene, and pyridine, are likely to be active against a number of pathogenic microorganisms, such as *Enterococcus faecalis*, *Mycobacterium tuberculosis*, *Mycobacterium BCG*, *Dialister invisus*, etc., including resistant strains. Furthermore, calculations show that complexation with Au^3+^ ions increases the probability of being active for all ligands. Despite the limitations inherent in the method [34], the results of preliminary screening can serve as a basis for the choice of ligands for research.

The study of binding between Au^3+^ complexes and proteins and DNA is often conducted [3,4,5,6,9,15,16,17,18,19,20,35,36]. However, in the referenced papers [3,4,5,6,9,15,16,17,18,19,20,35,36], the possible dissociation of the metal complex and separate binding of the ligand and cation to the biomolecule are not taken into account (despite the process of dissociation of metal complexes in the presence of DNA or protein being possible and abundant [30,37,38]). Furthermore, if enzymes such as thioredoxin reductase are considered biological targets, the dissociation of the Au^3+^ complex is necessary for inhibition, because the gold(III) ion binds to the residues of (seleno)cysteine with the formation of Au-S (Se) bonds. Therefore, the assessment of complex stability in the aqueous solution is a necessary step on the path towards the development of potential antibacterial drugs.

Due to the reasons above, the present contribution aims to study the stability of gold(III) complexes with five hydrazones derived from pyridoxal 5′-phosphate (PLP) in aqueous solution. The structures of the proposed ligands are given in Figure 1.

## 2. Results and Discussion

Gold(III) has complex chemistry in aqueous solution. The free cation readily undergoes hydrolysis, and exists as Au(OH)_2_^+^ even in 1 to 8 M strong acids [39]. Due to this, gold complexation is more convenient to study using its tetrachloride complex as a source of Au^3+^ ions (see, e.g., [40]). Therefore, the complexation of tetrachloroaurate(III) is a reaction of the ligand substitution. It should also be noted that chlorides can be substituted with OH^−^ ions, when the reaction transfers from an acidic medium to an alkali solution [41]. I. V. Mironov and L. D. Tsvelodub had shown that gold(III) tends to form very stable complexes with the chelating compounds, such as ethylenediamine and diethylenetriamine [40]; however, it was unknown whether hydrazones derived from pyridoxal 5′-phosphate could chelate Au^3+^. The study of any metal’s complexation with hydrazones is additionally complicated by the multiple protolytic equilibria of ligands, the poor solubility of free hydrazones and their complexes in the neutral or acidic media, and the large number of possible coordination compounds that can be formed under certain experimental conditions [42].

In the mass spectrum of the complex [Au(OH)(PLP-INH)] precipitated from an aqueous solution (Appendix A), one can find peaks with *m*/*z* = 581.7 and 604.1, corresponding to the complexes [Au + OH + L + H] (calcd. M = 581.05) and [Au + OH + L + H + Na] (calcd. M = 604.0).

Furthermore, the fluorescent emission spectra of free ligands differ from those of the mixture of Au^3+^ and ligands (Figure 2 and Appendix A), which is also indicative of the reaction that occurs between metals and hydrazones. The Stokes shift for all the studied hydrazones decreases by 25–45 nm, when cations are added. An interesting exception is the **PLP-F2H** hydrazone, whose luminescence intensity decreases greatly upon addition to the gold(III) solution (Figure 2b). The bimodal emission spectrum of **PLP-INH** hydrazone added to the gold(III) solution is also noteworthy (Appendix A). The decrease in the Stokes shift might be indicative of the Excited State Intramolecular Proton Transfer-off (ESIPT-off) mechanism caused by the complexation [43]. In other words, the free labile ligand capable of keto-enol tautomerism (as in most *o*-hydroxyl hydrazones and Schiff bases [44]) rigidifies upon complexation; the rotational freedom is limited, and the hydrazone molecule takes a less fluorescent configuration.

The UV–Vis spectra of free ligands are typical for hydrazones, showing intense absorbance at 295–315 nm. This peak refers to the electron transfer of the π-π-p-π conjugated system uniting the hydrazone molecules [29,32]. A long-wave shoulder peak at 340 nm should be presumably assigned to n-π* transfer.

The changes in UV–Vis spectra observed during the titration of Au^3+^ solution against hydrazones are less indicative of complex formation (Figure 3a,b and Appendix A) than the emission spectra. However, it should also be noted that the titration of gold(III) against hydrazone is different from that of only acid with the same C(H^+^) as the HAuCl_4_ solution (Figure 3). Namely, the changes associated with the formation of deprotonated species of hydrazones begin later, when the mixture of Au + H is titrated, than in the case of merely acid titration. This shows that the gold cation competes with the proton for the ligand molecule.

The pH value varied from 3.3 to 8.5 during the titration. It should be noted that in some cases (e.g., upon titration with **PLP-F2H** or **PLP-T3H**), at the end of the experiment, the appearance of a precipitated product in the solution was detected. It prevented us from further exploring the alkali region.

The results of spectrophotometric titration (Figure 3a,b and Appendix A) can be used to calculate the stability constants of the gold(III) complexes formed. The following processes of the general formula mH + nL + pAu + qOH = Au_p_(OH)_q_L_n_H_m_ should be set in the stoichiometric scheme for calculations of stability constants using KEV software [45] ([AuCl_4_]^−^ is denoted Au for convenience; the charges of other components are also omitted; note that all reactions involving ‘Au’ listed below are reactions of chloride substitution):H + L = HL(1)
2H + L = H_2_L(2)
3H + L = H_3_L(3)
Au + OH = AuOH(4)
Au + 2OH = Au(OH)_2_(5)
Au + 3OH = Au(OH)_3_(6)
Au + 4OH = Au(OH)_4_(7)
H + OH = HOH(8)
Au + L = AuL(9)
Au + H + L = AuHL(10)
Au + 2H + L = AuH_2_L(11)

The constants of the processes (1)–(3) are adopted from [46] for PLP-INH (log β_1_ = 11.37; log β_2_ = 19.60; log β_3_ = 23.81), PLP-F2H (log β_1_ = 11.43; log β_2_ = 19.72; log β_3_ = 23.79), PLP-T2H (log β_1_ = 11.47; log β_2_ = 19.75; log β_3_ = 24.24), PLP-T3H (log β_1_ = 11.48; log β_2_ = 19.31; log β_3_ = 24.15). The protonation constants of the PLP-F3H hydrazone are unknown; however, we took them to be equal to those of PLP-F2H, taking into account the closeness of the log β_i_ values determined for different hydrazones to each other. The constants of the processes (4)–(7) are taken from paper [41] (log β_(4)_ = 7.87; log β_(5)_ = 14.79; log β_(6)_ = 20.92; log β_(7)_ = 25.98). The value of log β_(8)_ = 13.91 was taken from the report [47]. The values of log β (Equations (9)–(11)) are to be determined.

We tested different stoichiometric models to find the most suitable one. They included the formation of a single complex AuH_m_L (m = 0–2), two complexes AuH_m_L and AuH_m−1_L (m = 0–2), and three complexes (Equations (9)–(11)). The latter gives the best fit of the calculated absorbance values to the experimental ones. It should be noted that a hydrazone substitutes three of four chloride anions in the coordination sphere; the last chloride likely remains or is substituted by the hydroxyl anion at high pH values (this possibility is taken into account in the model Equations (4)–(7)). The calculated stability constants, as well as the stepwise protonation constants log K_b1_ and log K_b2_ of the complexes (AuL + H = AuHL; AuHL + H = AuH_2_L, respectively) and the constants of gold(III) binding by the protonated ligands log K_f1_ and log K_f2_ (Au + HL = AuHL; Au + H_2_L = AuH_2_L, respectively), are given in Table 1.
log K_b1_ = log β(AuHL) − log β(AuL)(12)
log K_b2_ = log β(AuH_2_L) − log β(AuHL)(13)
log K_f1_ = log β(AuHL) − log β(HL)(14)
log K_f1_ = log β(AuH_2_L) − log β(H_2_L)(15)

It is interesting that hydrazones, despite being tridentate ligands, form less stable complexes than bidentate ethylenediamine [40]. This can be explained by the steric hindrance caused by the necessity of hydrazone molecule arrangement around the large Au(III) cation.

UV–Vis spectra of individual complex species (AuL, AuHL, AuH_2_L) were also calculated (Figure 4 and Appendix A). They explain why the changes in the UV–Vis spectra during spectrophotometric titration were not significant: the spectra of the complexes are similar to those of the protonated species. The similarity of the spectra, in turn, can be explained by the weak interaction of Au atomic orbitals (possessing high energy) with the low-energy molecular orbitals of ligands. For this reason, there is no band of charge transfer (ligand to metal or metal to ligand) in the UV–Vis spectra, which is typical for d-metal complexes with hydrazones, as studied previously [28,30]. Further quantum chemical calculations are required to confirm this assumption.

The relatively high values of the stability constant errors (Table 1) are the consequence of the difficulties experienced while studying the complexation of hydrazones, as well as the similarity of the spectra of protonated and complex species. They are characteristic for such systems, as was noted by the authors [42].

Finally, let us plot the speciation diagrams for all the studied complexes for the case C(HAuCl_4_) = C(L) = 0.0001 mol L^−1^ (Figure 5).

For every studied hydrazone, the following complex species of gold(III) are predominant at biologically relevant pH values (7.0–7.4): AuL, AuHL, AuOH, Au(OH)_2_. Hydrazone PLP-T2H shows the best binding chelating ability toward Au^3+^ ions. This can be explained by the possible additional stabilization of the complex due to the bond between the heterocyclic S atom and Au cation.

## 3. Materials and Methods

### 3.1. Chemicals

HAuCl_4_ (Aurat, Moscow, Russia) with purity of 99.9%, as claimed by the manufacturer, was used without additional purification. The synthesis of **PLP-INH**, **PLP-F2H**, **PLP-T2H**, **PLP-F3H**, **PLP-T3H** was performed as described in previous papers [29,32,48] from PLP and the corresponding hydrazones. HClO_4_ and NaOH (Reakhim, Russia) were standardized titrimetrically prior to use. In particular, perchloric acid was titrated against twice-recrystallized Na_2_B_4_O_7_·10H_2_O in the presence of methyl orange (indicator). This acid was further used for the titration of NaOH with two indicators (phenolphthalein and methyl orange) to control the traces of Na_2_CO_3_. The sodium carbonate did not exceed 2% compared to the NaOH concentration.

### 3.2. Methods

Spectrophotometric titrations were carried out using a double-beamed spectrophotometer, UV1800 (Shimadzu, Columbia, MD, USA), equipped with an external thermostat maintaining the temperature with an error of ±0.1 K. UV–Vis spectra were registered in the wavelength interval of 200–500 nm (error of wavelength determination was ±0.5 nm) and in the absorbance range of 0–2.5 units (the error of absorbance measurement ±0.004 units). Water was used as a blank solution. Standard quartz cells with an optical path length of 1 cm were used.

The main experiments included twenty additions (each of 10 μL) of aqueous solution of 1.5·10^−3^ mol L^−1^ hydrazones and 0.01074 mol L^−1^ NaOH to 2.7 mL of 2·10^−4^ mol L^−1^ AuCl_4_^−^ and 5·10^−4^ mol L^−1^ H^+^. The pH value during the titration varied within the range of 3.3 to 8.5. The ionic strength value set by HClO_4_ varied from 0.001 to 0.002 mol L^−1^ during titration. Stability constants and speciation diagrams were calculated using the KEV software [45]. See the Results and Discussion Section for details of equilibria in the stoichiometric model (Equations (1)–(11)).

The emission fluorescent spectra were registered using an RF6000 setup (Shimadzu, Columbia, MD, USA) at the excitation wavelength λ_ex_ = 365 nm, in the emission wavelength range 390–700 nm, at a scanning rate of 6000 nm/min. Excitation and emission slit widths were set at 5 nm. Standard quartz cells with an optical path length of 1 cm were used. The concentrations of ligands, metals, and H^+^ were equal to those at the end of UV–Vis titration.

Mass spectra (MALDI TOF) were recorded using a Shimadzu Biotech Axima Confidence setup (Shimadzu, Columbia, MD, USA).

The following definitions were used for the equilibrium constants:(1)The symbol β refers to the total equilibrium constant of the reaction with the general formula mH + nL + pAu + qOH = Au_p_(OH)_q_L_n_H_m_;(2)The symbol K_bm_ refers to the stepwise protonation constant of the process with the general formula AuLH_m-1_ + H = AuLH_m_;(3)The symbol K_fm_ refers to the equilibrium constant of the reaction of gold(III) binding to the protonated ligands with the general formula Au + H_m_L = AuLH_m_.

## 4. Conclusions

The complexation of gold(III) taken as tetrachloroaurate with hydrazones derived from pyridoxal 5′-phosphate was studied in aqueous solution at *T* = 298.2 K. Complexes of AuL, AuHL, AuH_2_L stoichiometry were found to be formed under the applied experimental conditions. Complex formation slightly (except for the hydrazone derived from PLP and 2-furoylhydrazide) quenched the intrinsic fluorescence of the ligands and decreased the Stokes shift. The stability constants were determined for complexes of different stoichiometry. Analysis of the speciation diagrams has shown that the predominate complex species of gold(III) in the physiological range of pH values are AuL, AuHL, AuOH, and Au(OH)_2_.

The present contribution is hopefully the first step on the path to studying the coordination chemistry of gold(III) in aqueous solution regarding its further use in medicine. The following topics should be addressed and questions answered (as in our future papers):Why are the UV–Vis spectra of gold(III) complexes with hydrazones similar to those of free hydrazones? The high-level quantum chemical calculations might be of great use to answer this question.Because gold(III) is not completely bound to the complex with hydrazone, the study of the interactions between gold(III) (in the form of a tetrachloroaurate or mixed chloride and hydroxyl complex) and proteins or DNA is in order.Would the gold(III) complexes with hydrazones dissociate completely in the presence of proteins or nucleic acid?Would gold(III) complexes with hydrazones derived from pyridoxal 5′-phosphate have any effect against pathogenic microorganisms?

## Figures and Tables

**Figure 1 molecules-27-07346-f001:**
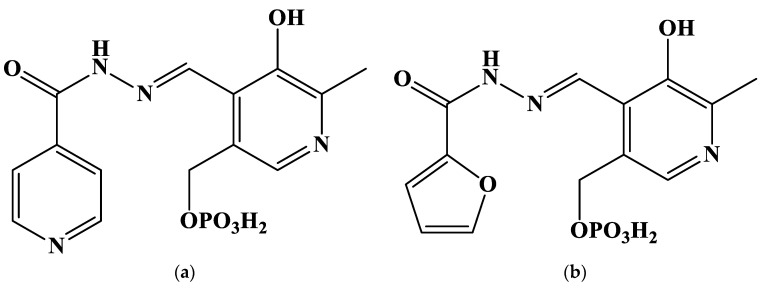
Structures of the studied hydrazones derived from pyridoxal 5′-phosphate and (**a**) isoniazid (**PLP-INH**); (**b**) furoyl-2-hydrazide (**PLP-F2H**); (**c**) thiophene-2-carbohydrazide (**PLP-T2H**); (**d**) 2-methylfuroyl-3-hydrazide (**PLP-F3H**); (**e**) thiophene-3-carbohydrazide (**PLP-T3H**).

**Figure 2 molecules-27-07346-f002:**
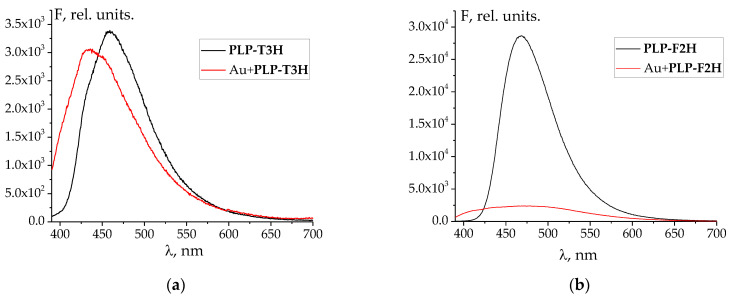
Emission spectra of free hydrazones and their mixtures with Au^3+^: (**a**) **PLP-T3H**; (**b**) **PLP-F2H**. λ_ex_ = 365 nm. Concentrations of ligands, metals, and H^+^ are equal to those at the end of UV–Vis titration (see below).

**Figure 3 molecules-27-07346-f003:**
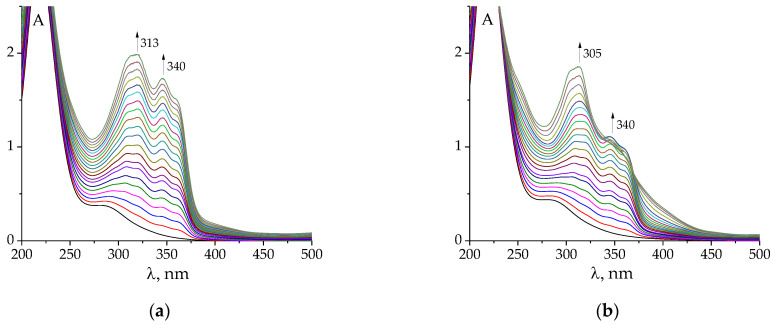
Examples of changes in UV–Vis spectra of HAuCl_4_ (**a**,**c**) and HClO_4_ (**b**,**d**) after the addition of **PLP-F2H** (**a**,**c**), **PLP-T2H** (**b**,**d**). Titrant: C(AuCl_4_^−^) = 2·10^−4^ mol L^−1^; C(H^+^) = 5·10^−4^ mol L^−1^ (**a**,**c**); C(H^+^) = 5·10^−4^ mol L^−1^ (**b**,**d**). Titrant: C(**PLP-F2H**) = 1.5·10^−3^ mol L^−1^; C(OH^−^) = 0.01074 mol L^−1^ (**a**,**c**); Titrant: C(**PLP-T2H**) = 1.5·10^−3^ mol L^−1^; C(OH^−^) = 0.01074 mol L^−1^ (**b**,**d**). Twenty additions of 10 μL volume.

**Figure 4 molecules-27-07346-f004:**
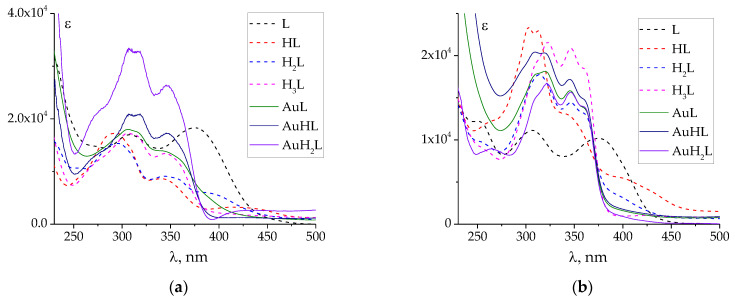
Calculated UV–Vis spectra of individual protonated and complex species of **PLP-INH** (**a**) and **PLP-F2H** (**b**). The spectra of protonated species are adopted from [44]. ‘Au’ denotes tetrachloroaurate(III).

**Figure 5 molecules-27-07346-f005:**
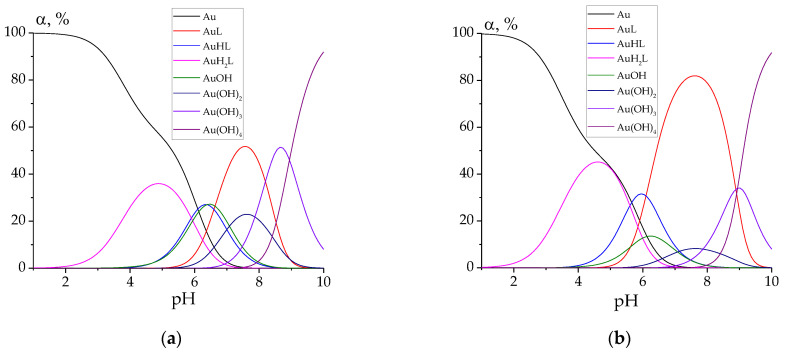
Speciation diagrams for the solutions containing 0.0001 mol L^−1^ of HAuCl_4_ and 0.0001 mol L^−1^ of hydrazones: **PLP-INH** (**a**); **PLP-F2H** (**b**); **PLP-F3H** (**c**); **PLP-T2H** (**d**); **PLP-T3H** (**e**). ‘Au’ denotes tetrachloroaurate.

**Table 1 molecules-27-07346-t001:** Stability constants of gold(III) complexes with hydrazones derived from pyridoxal 5′-phosphate in aqueous solution at *T* = 298.2 K, *I* ~ 0.

Hydrazone	PLP-INH	PLP-F2H	PLP-F3H	PLP-T2H	PLP-T3H
log β (AuL) ^1^	11.2 ± 0.5	12.4 ± 0.9	12.0 ± 0.5	13.1 ± 0.8	12.5 ± 1.0
log β (AuHL) ^1^	17.8 ± 0.8	18.5 ± 0.6	18.6 ± 0.6	20.3 ± 0.6	18.4 ± 0.3
log β (AuH_2_L) ^1^	23.7 ± 0.9	24.1 ± 0.7	24.7 ± 0.7	26.0 ± 0.5	24.2 ± 0.2
log K_b1_ ^2^	6.6 ± 0.9	6.1 ± 1.1	6.6 ± 0.8	7.2 ± 1.0	5.9 ± 1.0
log K_b2_ ^2^	5.9 ± 1.2	5.6 ± 0.9	6.1 ± 0.9	5.7 ± 0.8	5.8 ± 0.4
log K_f1_ ^3^	6.4 ± 0.9	7.1 ± 0.6	7.2 ± 0.6	8.8 ± 0.6	6.9 ± 1.0
logK_f2_ ^3^	4.1 ± 0.9	4.4 ± 0.7	5.0 ± 0.8	6.2 ± 0.7	4.9 ± 0.4

^1^ The errors are the half-widths of the confidence interval at a confidence probability of 0.95 and sample size of 4 to 6 experiments. ^2^ The errors are the square roots taken from the sum of squared errors of log β(AuH_x_L) used for calculations. ^3^ The errors are the square roots taken from the sum of squared errors of log β(AuH_x_L) and protonation constants of hydrazones used for calculations.

## Data Availability

Not applicable.

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
