# Peer review of "Complexation of Gold(III) with Pyridoxal 5′-Phosphate-Derived Hydrazones in Aqueous Solution"

_molecules, 2022, doi:10.3390/molecules27217346_

Round 1
Reviewer 1 Report
Revision:
Complexation of Gold(III) with Pyridoxal 5’-phsphate derived hydrazones in aqueous solution.
This paper reports a very interesting investigation on the speciation of Gold(III) in the presence of some Pyridoxal 5’-phsphate derived hydrazones used for different diseases.
Investigation were carried out by using some different techniques, even if the results on the speciation studies were obtained from UV-Vis. A very simple speciation model, characterized by only three MHiL species, was obtained with a stability constant for the ML species ranging from ~12 to 13.1.
Comments:
11. Page 3 rows 89-90: authors report that hydrazones derived from pyridoxal 5’-phosphate are characterized by poor solubility; can authors report if solubility measurements on these ligands has been carried out, and if yes, to report some data. As an example footnote of Figure 2 reports that a C(PLP-F2H) = 1.5 10-3 mol L-1 has been used for UV-Vis titrations.
22. page 4 row 103: it is useful for the readers to define: ESIPT-off mechanism
33. page 5 rows 126-136; authors report the equilibria of the species involved in the speciation model; it is better if they report only a general equilibrium of formation;
along the manuscript authors report the symbols log b; pK; log K; it is useful for readers less confident with the studies of equilibria to define they.
55. page 6 rows 137-144: authors report the overall protonation constants of the ligand here investigated, except for PLP-F3H; why the protonation constants of this last ligand are unknown?
66. Page 6 row 144: authors report log b9-11: I assume that it refers to the protonation constants reported in the equations (9-11). it must be corrected.
77. Page 6 rows 146-154: authors report that different speciation models were tested and that the proposed one has been selected on the basis of the best fit of the calculated absorbance. Generally some rules were used to selected the best speciation model, bases on the ration of the variance between the tested models and the selected one; simplicity of the model; formation percentage of the proposed species; comparison with similar systems. Can authors justify their procedure?
88. Page 6 row 156-159: authors report the general equation to calculated the stepwise formation constants. Can Authors explain the utility to calculate in two different ways the stepwise constants?
99. Page 6 Table 1: authors reports the overall and stepwise formation constants of the complexes; for the equations 12-13 of page 6, the values are expressed as pKi values; this is not correct since if they use this formalism, the formation constants must have negative values: -6.6; -6.1; -6.6, etc. Inf act, in general pKi = - log Ki. The stepwise formation constants expressed by means of the equation 12-13, are log Ki value calculated from the difference of the log bi constants and another one.
110. Page 6 Table 1: results are expressed at T = 298.2 K and I~0; since the data reported in this table, are experimental values and not formation constants obtained from the extrapolation at I à 0, it is incorrect to use the symbol log b0 (but also for pK0 and log K0f), that in each case, is not defined in the text.
111. Since the investigated systems have biological interest, why studied were not carried out at T = 310.15 K and in ionic medium that can simulate biological fluid?, namely NaCl 0.15 mol L-1. Another very interesting idea is to carry these studies in synthetic blood plasma.
112. Stability constants reported in Table 1 have high errors; at page 7 rows 185-188, they justify this: in my opinion the high errors of the formation constants can be due to the not correct speciation model or that some other species must be considered; to better understand this problem, it would be useful to report a graph containing both the experimental curves and the calculated ones.
113. Page 8 graphs 4. The distribution diagrams were drawn up to pH = 10, but in the text, authors reports that investigations were done up to pH = 8.5. Can authors justify this, and why experimental investigation were stopped at pH = 8.5. The formation of precipitate was observed).
114. Page 9 rows 205-206: it is important to specify how the chemicals were standardized.
115. Page 9 row 215: 0.01074 NaOH; correct as 0.01074 mol L-1 NaOH.
116. Page 9 row 217: authors report that the ionic strength was set by using HCLO4 varied from 0.001 to 0.002 mol L-1. This concept is not correct since if we want to work at constant ionic strength, it is important to add a strong electrolyte at a concentration higher with respect to the other component. In this case, it is important to consider, the dissociation of the ligands and the hydrolysis of the metal along the titration. So in my opinion, the consideration of the authors are not correct and the ionic strength of the solution unknown.
117. Page 9 rows 221-225: authors used the spectrofluorimetry to register the spectra and compare those of the free ligand with the mixture containing the metal. Why did authors not carry titration by using also this technique??? This investigation can help authors to understand why formation constants have so high error.
118. Similarly, why they used MASS SPECTROSCOPY to investigate the solid state, and they do not carry investigation of the solution at different pH values?
Considering the comment above reported, I suggest the publication of the present manuscript, only after a major revision.
Reviewer 2 Report
The manuscript “Complexation of Gold(III) with Pyridoxal 5’-Phosphate Derived Hydrazones in Aqueous Solution” describes a well-detailed study concerning the development of new antibacterial drugs based on the Au3+ complexes. The authors reported the investigation of gold(III) complexation with five hydrazones derived from the well-known biologically active 13 compounds, pyridoxal 5′-phosphate. The complex formation in aqueous solutions was confirmed by different spectroscopic techniques.
I can recommend acceptance for publication after addressing the following remarks:
1. In the introduction section, the rational design of novel molecules is not discussed by the authors.
2. Please cite the following articles in the introduction section. (doi.org/10.1016/j.molstruc.2021.130965; doi.org/10.1016/j.molstruc.2021.131933)
3. Authors should write the physical state of the synthesized compounds.
4. The explanation of the UV-Vis results of the studied compounds should be detailed.
5. Calculate the quantum yield of all the target compounds.
6. Attach the FTIR spectra of all the synthesized compounds in the Supplementary information file. Insert the following "vmax" symbol for all IR data as follows: FTIR (KBr, cm-1) vmax followed by IR absorbances in the text.
7. Authors should include the structure-property relationship of the subjected compounds.
8. I would recommend that the manuscript should be checked completely for typographical errors.
Reviewer 3 Report
I have reviewed manuscript number": molecules-1995726, entitled “Complexation of Gold(III) with Pyridoxal 5′-Phosphate Derived Hydrazones in Aqueous Solution”. The authors studied the stability of gold(III) complexes with five hydrazones derived from pyridoxal 5′-phosphate (PLP) in the aqueous solution. They also explored the stoichiometric composition of the complexes formed and their stability constants determined using the UV-Vis titration method. Overall, the manuscript is written and organized well. After checking the whole manuscript, some comments can be seen:
1. Why is the research important to do?
2. What properties make the studied compounds peculiar?
3. The authors must be added the following references in the hydrazone of the introduction section:
https://doi.org/10.1016/j.ica.2021.120535
https://doi.org/10.1016/j.molstruc.2021.131691
4. Authors should comment on the solubility nature of the synthetic compounds in the revised manuscript.
5. Some “References” should be corrected based on the “Molecules” format.
6. The English language of the manuscript should be carefully checked and some typos should be corrected.
After the authors correct all the above issues, this paper can be published after minor revision.
Round 2
Reviewer 1 Report
Taking into account the revision process of the manuscript, I suggest its publication in Molecules